# PrACTiS: Perceiver-Attentional Copulas for Time Series

## Abstract

Transformers incorporating copula structures have demonstrated remarkable performance in time series prediction. However, their heavy reliance on self-attention mechanisms demands substantial computational resources, thus limiting their practical utility across a wide range of tasks. In this work, we present a model that combines the perceiver architecture with a copula structure to enhance time-series forecasting. By leveraging the perceiver as the encoder, we efficiently transform complex, high-dimensional, multimodal data into a compact latent space, thereby significantly reducing computational demands. To further reduce complexity, we introduce midpoint inference and local attention mechanisms, enabling the model to capture dependencies within imputed samples effectively. Subsequently, we deploy the copula-based attention and output variance testing mechanism to capture the joint distribution of missing data, while simultaneously mitigating error propagation during prediction. Our experimental results on the unimodal and multimodal benchmarks showcase a consistent 20% improvement over the state-of-the-art methods, while utilizing less than half of available memory resources.

## 1 Introduction

Time-series prediction remains an enduring challenge since it requires effectively capturing global patterns (e.g., month-long trends) and localized details (e.g., abrupt disruptions). This challenge becomes particularly pronounced when dealing with non-synchronized, incomplete, high-dimensional, or multimodal input data. For instance, consider a time series consisting of $N$ regularly-sampled and synchronously-measured values, where measurements are taken at intervals of length $T$. If the time-step is unobserved at rate $r$, then there are $(1-r)NT$ observed values that are relevant for inference. Consider an asynchronously-measured time series, where input variables are observed at different times, resulting in each time-step having only $1/N$ of its variables observed. In this scenario, only $(1-r)T$ values remain relevant for inference within the time series. Consequently, employing a synchronous model to address non-synchronized time series results in a missingness rate of $(N-1)/N$. This missingness rate grows rapidly as the number of variables increases, reaching 95% with just 20 variables in the time series. When designing an architecture to handle missing data, it is crucial to utilize techniques for approximating missing values while ensuring the computational overhead does not exceed the effort required to extract valuable insights from the observed data. To this end, TACTiS (Drouin et al., 2022) presents an attention-based model (Vaswani et al., 2017) tailored for time series. This model tokenizes input variables and utilizes a transformer-based encoding and decoding approach, making it a suitable choice for modeling non-synchronized time series data. Tokenization also offers significant advantages for missing data, as unobserved data can be seamlessly excluded from the token stream. Additionally, TACTiS leverages a copula structure (Nelsen, 2006) as a flexible model to represent the sequence distribution and achieves remarkable prediction performance. Particularly, it represents the joint distribution with a non-parametric copula that is a product of conditional probabilities. To ensure that the product give a valid copula, TACTiS considers permutations of the margins during training such that a level of permutation invariance occurs. This, however, yields an exchangeable class of copulas in the limit of infinite permutations, diminishing the utility of the non-parameteric copula. In addition, the transformer architecture in TACTiS poses significant computational demands related to the self-attention mechanism.

In this paper, we introduce the Perceiver-Attentional Copulas for Time Series (PrACTiS) architecture, a new approach that combines the Perceiver IO model (Jaegle et al., 2021a) with attention-

based copulas to enhance time series modeling and address computational efficiency challenges. Our model consists of the perceiver-based encoder and the copula-based decoder, enabling the incorporation of a more general class of copulas that are not exchangeable. Specifically, the class of copulas in PrACTiS are the *factor copulas*, which are conditionally exchangeable based on the factor. Initially, PrACTiS transforms the input variables into temporal embeddings through a combination of input embedding and positional encoding procedures. In this phase, the observed and the missing data points are encoded (i.e., the value of missing data points are masked). Subsequently, PrACTiS utilizes a *latent* attention-based mechanism inspired by the perceiver architecture. This mechanism efficiently maps the acquired input embeddings to a lower-dimensional latent space. Since all subsequent computations are performed within this compact latent space, it helps reduce the complexity from a quadratic to a sub-quadratic level. Lastly, the decoder leverages the copula structure to formulate the joint distribution of missing data using latent embeddings. This distribution undergoes a sampling process to yield the predicted outcomes. Our proposed model can effectively handle synchronized, non-synchronized, and multimodal data, expanding its applicability to diverse domains.

To validate the efficacy of PrACTiS, we conduct extensive experiments on the unimodal datasets from the Monash Time Series Forecasting Repository (Godahewa et al., 2021) (i.e., electricity, traffic, and fred-md) and the multimodal datasets, such as room occupation (Candanedo, 2016), interstate traffic (Hogue, 2019), and air quality (Chen, 2019). We also conduct memory consumption scaling experiments using random walk data to demonstrate the memory efficiency of PrACTiS. The results demonstrate the competitive performance of our model compared to the state-of-the-art methods, including TACTiS, GPVar (Salinas et al., 2019), SSAE-LSTM (Zhu et al., 2021), and deep autoregressive AR (Kalliovirta et al., 2015) while utilizing as little as 10% of the memory resources.

## 2 RELATED WORK

Neural networks for time series forecasting (Zhang et al., 1998) have undergone extensive research and delivered impressive results when compared to classical statistical methods (Box et al., 2015; Hyndman et al., 2008; Yanchenko & Mukherjee, 2020). Notably, both convolutional (Chen et al., 2020) and recurrent neural networks (Connor et al., 1994; Shih et al., 2019; Hochreiter & Schmidhuber, 1997) have demonstrated the power of deep neural networks in learning historical patterns and leveraging this knowledge for precise predictions of future data points. Subsequently, various deep learning techniques have been proposed to address the modeling of regularly-sampled time series data (Oreshkin et al., 2019; Le Guen & Thome, 2020; de Bézenac et al., 2020; Lim & Zohren, 2021; Benidis et al., 2022). Most recently, the transformer architecture, initially designed for sequence modeling tasks, has been adopted extensively for time series forecasting (Li et al., 2019; Lim et al., 2021; Müller et al., 2021). Using the properties of the attention mechanism, these models excel at capturing long-term dependencies within the data, achieving remarkable results. In addition to these developments, score-based diffusion models (Tashiro et al., 2021) achieved competitive performance in forecasting tasks. However, it is worth noting that the majority of these approaches are tailored for handling regularly sampled and synchronized time series data. Consequently, they may not be optimal when applied to non-synchronized datasets. In financial forecasting, the copula emerges as a formidable tool for estimating multivariate distributions (Aas et al., 2009; Patton, 2012; Krupskii & Joe, 2020; Größer & Okhrin, 2022; Mayer & Wied, 2023). Its computational efficiency has led to its use in the domain adaptation contexts (Lopez-Paz et al., 2012). Moreover, the copula structure has found utility in time series prediction when coupled with neural architectures like LSTMs (Lopez-Paz et al., 2012) and the transformer (Drouin et al., 2022), enabling the modeling of irregularly sampled time series data. While previous research has explored non-synchronized methods (Chapados & Bengio, 2007; Shukla & Marlin, 2021), their practicality often falters due to computational challenges. The TACTiS (Drouin et al., 2022) method combines the transformer architecture with copulas, and it achieved significant advancements over existing models, making the approach applicable to both synchronized and non-synchronized datasets. Nonetheless, it is important to note that the inherent computational overhead associated with the transformer's self-attention mechanism poses limitations, particularly when applied to high-dimensional inputs such as multimodal data. To mitigate this computational complexity, we propose the adoption of the perceiver IO (Jaegle et al., 2021b;a) as the encoder, paired with a copula-based decoder. In addition, we utilize the midpoint inference (Liu et al., 2019) during the decoding phase of the model. This approach restricts conditioning and effectively embodies a form of sparse attention (Child et al., 2019; Tay et al., 2020; Roy et al., 2021), although the sparsity pattern is determined through a gap-filling process.

## 3 PRELIMINARIES

TACTiS (Drouin et al., 2022) has exhibited outstanding performance in the domain of time series prediction. It harnesses the self-attention mechanism (Vaswani et al., 2017) at two critical stages within its operation. Firstly, TACTiS employs self-attention to encode input time series variables, effectively transforming them into a sequence of generalized tokens. This transformation enables the model to process and analyze the temporal aspects of the data efficiently. Secondly, it applies self-attention over this resulting token sequence to generate a conditional distribution of inferred variables using a parameterized copula. An innovative aspect of TACTiS involves permuting modeled variables, allowing the model to determine the optimal ordering of these conditional distributions. Furthermore, it introduces a stochastic element by randomly selecting permutations for each inference sample, accommodating scenarios where a fixed order may not be ideal. This feature grants TACTiS a competitive edge in modeling asynchronous time series data. These features allow TACTiS to accel in missing data inference. It accomplishes this by removing unobserved data from the token stream during tokenization, allowing the model to generate precise predictions even when dealing with incomplete input.

Let $\mathcal{X}$ denote the a time series of interest, $\mathcal{X} = \{X_1, X_2, \ldots, X_i, \ldots\}$. Each element $X_i$ is defined as a quadruple: $X_i = (v_i, c_i, t_i, m_i)$, where $v_i$ is the value, $c_i$ is an index identifying the variable, $t_i$ is a time stamp, and $m_i$ is a mask indicating whether the data point is observed (i.e., with available value) or needs to be inferred (i.e., missing data). For synchronously measured time series data, we can organize it into a data matrix denoted as $X_{c,t}$. This matrix has rows corresponding to individual variables and columns corresponding to different timestamps when measurements were recorded. The TACTiS architecture comprises an encoder and a decoder. First, the encoder generates embeddings for each data point, $\vec{x}_i$, which includes the value $v_i$, a learned encoding for the variable $c_i$, an additive sinusoidal positional encoding indicating the position of $t_i$ within the overall time series, and the mask $m_i$. Subsequently, TACTiS employs a self-attention mechanism represented as $\texttt{self\_attention}(\texttt{K}, \texttt{Q}, \texttt{V})$, where $K$ represents keys, $Q$ is a query, and $V$ is a set of values. By utilizing learned functions for generating keys and values, $\texttt{key\_encode}()$ and $\texttt{value\_encode}()$, TACTiS derives a tokenized representation, denoted as $\vec{z}_i$, for each data point. This is achieved by passing the input embeddings through a stack of residual layers, as follows:

$$\vec{z}_i = \texttt{self\_attention}(\texttt{key\_encode}(\vec{x}_{\neg i}), \vec{x}_i, \texttt{value\_encode}(\vec{x}_{\neg i}))$$

Next, the decoder is specifically designed to learn the joint distribution of the missing data points conditioned on the observed ones. To achieve this, the attention-based decoder is trained to mimic a non-parametric copula (Nelsen, 2006). Let $x^{(m)}$ and $x^{(o)}$ represent the missing and observed data points, respectively. Let $F_i$ be the $i^{\text{th}}$ marginal cumulative distribution function (CDF) and $f_i$ be the marginal probability density function (PDF). The copulas, under Sklar's theorem (Sklar, 1959), allow for separate modeling of the joint distribution and the marginals, which has particular relevance to the case of sequence modeling. To model the marginal CDF, TACTiS employs a normalizing flow technique known as Deep Sigmoidal Flow (Huang et al., 2018). The marginal PDF is obtained by differentiating the marginal CDF. The copula-based structure $g_\phi$ is described as follows:

$$g_\phi\left(x_1^{(m)}, \ldots, x_{n_m}^{(m)}\right) = c_{\phi_c}\left(F_{\phi_1}\left(x_1^{(m)}\right), \ldots, F_{\phi_{n_m}}\left(x_{n_m}^{(m)}\right)\right) \times f_{\phi_1}\left(x_1^{(m)}\right) \times \ldots \times f_{\phi_{n_m}}\left(x_{n_m}^{(m)}\right),$$

where $c_{\phi_c}$ is the density of a copula, and $c_{\phi_c}(F_{\phi_1}(x_1^{(m)}), \ldots, F_{\phi_{n_m}}(x_{n_m}^{(m)})) = c_{\phi_{c1}}(F_{\phi_1}(x_1^{(m)})) \times c_{\phi_{c2}}(F_{\phi_2}(x_2^{(m)}) \big| F_{\phi_1}(x_1^{(m)})) \times \ldots \times c_{\phi_{cn_m}}(F_{\phi_{n_m}}(x_{n_m}^{(m)}) \big| F_{\phi_1}(x_1^{(m)}), \ldots, F_{\phi_{n_m-1}}(x_{n_m-1}^{(m)}))$.

During the decoding phase, TACTiS selects a permutation, denoted as $\gamma$, from all data points, ensuring that observed data points come before those awaiting inference. It then utilizes the learned key and value functions, $\texttt{key\_decode}()$ and $\texttt{value\_decode}()$, to derive distributional parameters, $\theta_{\gamma(i)}$, for each datapoint awaiting inference as follows:

$$\theta_{\gamma(i)} = \texttt{self\_attention}(\texttt{key\_decode}(\vec{z}_{\gamma(j)<\gamma(i)}), \vec{z}_{\gamma(i)}, \texttt{value\_decode}(\vec{z}_{\gamma(j)<\gamma(i)}))$$

Finally, TACTiS introduces a parameterized diffeomorphism $f_{\phi,c} : (0, 1) \mapsto \mathbb{R}$. When $\theta$ represents the parameters for a distribution $p_\theta$ over the interval $(0, 1)$, TACTiS proceeds by either sampling data points as $\hat{x}_i = f_{\phi,c_i}(u_i), \quad u_i \sim p_{\theta_i}$, or computing the conditional likelihood: $p_{\theta_i}(f_{\phi,c_i}^{-1}(x_i))$.

It's important to note that both self-attention mechanisms within TACTiS involve pairwise calculations among the variables. Consequently, the encoder's computational complexity scales as $O(N^2)$, where $N$ is the number of data points in the time series. Conversely, the decoder's complexity scales as $O(S(S + H))$, where $S$ represents the number of data points to be inferred and $H$ denotes the number of observed data points. To address the computational complexity of the encoder in TACTiS, especially for synchronously measured time series, Drouin et al. (2022) have implemented a temporal transformer variant. This variant applies attention iteratively, first within a time step and then across time steps, effectively reducing the encoder's complexity to $O(n^2T + nT^2)$, where $n$ is the number of variables, and $T$ is approximately the number of time steps. However, it's worth noting that the self-attention-based decoder in TACTiS maintains its complexity scaling as $O(S(S + H))$.

Here, TACTiS uses a training procedure involving random permutations to establish an approximately valid copula. In particular, all margins are approximately the same and approximately uniform. Once a valid copula is obtained, it leverages Sklar's theorem to combine copula density with marginal densities for maximum likelihood estimation (MLE). However, the conditional copula factorization, as expressed by $c(u_1, \ldots, u_d) = c(u_{\pi_1}) \times c(u_{\pi_2} \mid u_{\pi_1}) \times \cdots \times c(u_{\pi_d} \mid u_{\pi_1}, \ldots, u_{\pi_{d-1}})$ for permutations of indices $\pi$, carries significant implications, especially in the asymptotic limit of infinite permutations. As the order of the marginals becomes irrelevant, the copula converges into a family of exchangeable copulas. Relying on exchangeability to ensure the validity of a copula undermines the potential advantages of utilizing a nonparametric copula, ultimately diminishing its expected benefits.

# 4 PRACTIS

We propose the integration of the Perceiver model (Jaegle et al., 2021b;a) as the encoder with the copula-based decoder, aiming to enhance the expressiveness of dependence between covariates, elevate prediction performance, and streamline the complexity of TACTiS. This integrated model, called *Perceiver-Attentional Copulas for Time Series* (PrACTiS), represents a groundbreaking advancement in generative modeling for time series data. PrACTiS utilizes the advantages of both the self-attention mechanism and latent-variable-based attention mechanisms from perceivers. Notably, it enables the modeling of dependencies between covariates, which can converge into a factor copula (Oh & Patton, 2017; Krupskii & Joe, 2013) described as follows:

$$C(u_1, \cdots, u_d) = \int_{[0,1]^k} \prod_{j=1}^{d} F_{j|Z_1, \cdots, Z_k}(u_j | z_1, \cdots, z_k) dz_1 \cdots z_k \qquad (1)$$

The factor copula is particularly well-suited for modeling high-dimensional data, primarily because it permits the specification of a parametric form with linear $O(n)$ dependency parameters, rather than the computationally burdensome $O(n^2)$ parameters, where $n$ is the number of observed variables. Furthermore, the factor copula model proves invaluable in cases where the interdependence among observed variables is contingent upon a limited number of unobserved variables, particularly in situations where there is a presence of tail asymmetry or tail dependence within the dataset. In numerous multivariate scenarios, the dependence on observed variables can be explained through latent variables. Importantly, this approach dispenses with the assumption of exchangeability, allowing the copula to adopt a more general form. The perceiver considers this structure conditioned on latent variables. As a result, our proposed PrACTiS model is capable of effectively handling multimodal input data, while significantly reducing computational complexity. The perceiver serves as the driving force behind PrACTiS, enabling it to efficiently process a wide spectrum of data types. The overview architecture of our proposed model is illustrated in Figure 1. Next, we discuss detailed components of the PrACTiS model.

## 4.1 PERCEIVER-BASED ENCODING

Initially, each data point $\vec{x}_i$ undergoes embedding via the input embedding and positional encoding processes. Subsequently, these embeddings are passed through the perceiver-based encoder. Here, the encoder leverages a predefined set of learned latent vectors $\vec{u}_k$ for the cross-attention mechanism `cross_attention(K, Q, V)`, where $K$ is a set of keys, $Q$ is a query, and $V$ is a set of values. Through the utilization of learned key and value-generating functions, `key_latent()` and

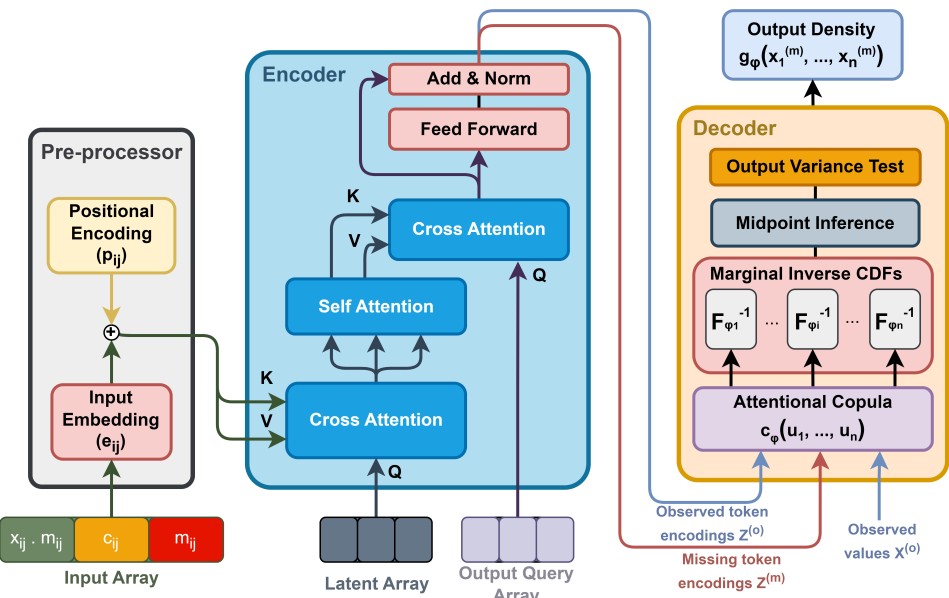

Figure 1: The overview architecture of PrACTiS. The pre-processor includes input embedding, and positional encoding layers to capture temporal dependencies in the input data. The encoder uses the cross-attention mechanisms to map the embedding to a lower-dimensional latent space. The decoder constructs the joint distribution of missing data using the copula-based structure.

`value_latent()`, the encoder derives latent vectors $\vec{w}_i$, which effectively encapsulate the temporal information through cross-attention with the set of observed vectors $\vec{X}_O$ as follows:

$$\vec{w}_k = \texttt{cross\_attention}(\texttt{key\_latent}(\vec{X}_O), \vec{u}_k, \texttt{value\_latent}(\vec{X}_O))$$

Following additional self-attention-based processing on the set of latent vectors, the perceiver-based encoder proceeds to employ cross-attention with the latent vector set $\vec{W}$, to generate tokens for each data point. This operation involves using the learned key, query, and value-generating functions, `key_encode()`, `query_encode()`, and `value_encode()`, to derive token vectors $\vec{z}_i$ as follows:

$$\vec{z}_i = \texttt{cross\_attention}(\texttt{key\_encode}(\vec{W}), \texttt{query\_encode}(\vec{x}_i), \texttt{value\_encode}(\vec{W}))$$

Aligned with the perceiver architecture, the number of latent features $K$ is intentionally maintained at a considerably smaller scale compared to the total number of data points $N$. This strategic choice serves to manage computational complexity, which scales at $O(NK)$. The initial cross-attention step in PrACTiS assumes a pivotal role by encoding a comprehensive global summary of the observed data from the time series into a set of concise latent vectors. These latent vectors effectively capture the essential information embedded within the entire dataset. Subsequently, PrACTiS generates tokens for each individual data point by efficiently querying relevant global information from the previously obtained latent summary in the second cross-attention step. This process ensures that each token encompasses vital contextual details drawn from the overall dataset, as necessitated.

## 4.2 MIDPOINT INFERENCE AND LOCAL ATTENTION

To enhance computational efficiency while maintaining the prediction performance, we propose the midpoint inference mechanism with temporally local self-attention to effectively reduce computational overhead. Instead of relying on random permutations to establish the conditioning structure, our method employs permutations that recursively infer midpoints within gaps in the observed data. When dealing with a continuous sequence of missing data points for the same variable, we determine the depth of each data point based on the number of midpoint inferences required within that sequence before considering the data point itself as a midpoint. Notably, observed data points are

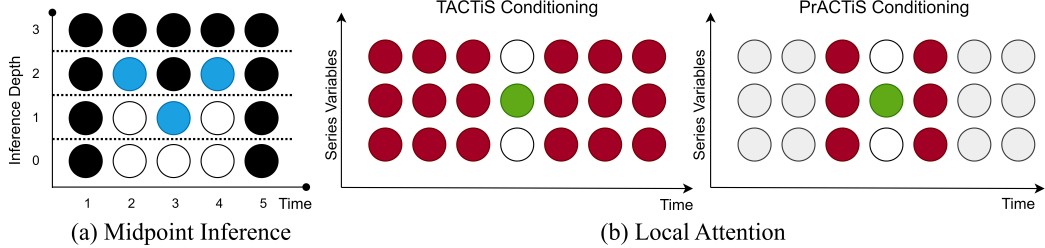

(a) Midpoint Inference                     (b) Local Attention

Figure 2: (a) Visualization of the midpoint inference mechanism: blue-filled points represent the points earmarked for inference at a particular depth, while black points represent those already observed or inferred at that depth and the white points are unobserved. (b) Comparison between the attention mechanisms in TACTiS and PrACTiS, both utilizing a local window containing only the nearest tokens: green-filled points indicate the currently sampled variable, while red points signify the variables to which the sampled token directs its attention during the sampling process.

assigned shallower depths compared to data points that are yet to be observed. Consequently, we sample a permutation $\gamma$ that positions data points with shallower assigned depths before those with deeper depths. Here, we determine midpoints by considering the number of data points between the prior observation and the next observation, as visually depicted in Figure 2. This method is well-suited for regularly or nearly-regularly sampled time series data. For each data token $\vec{z}_i$, our approach selects a set of conditioning tokens $\vec{H}_i$. These conditioning tokens comprise both past and future windows, consisting of the $k$ closest tokens for each variable in the series that precede $\vec{z}i$ within the generated permutation $\gamma$. Figure 2(b) illustrates the proposed local-attention conditioning mechanism in comparison with TACTiS's global self-attention. Here, PrACTiS employs learned key and value-generation functions, key_decode() and value_decode(), to derive distributional parameters $\theta\gamma(i)$ for each data point to be inferred, following the ordering imposed by $\gamma$ as follows:

$$\theta_i = \texttt{self\_attention}(\texttt{key\_decode}(\vec{H}_i), \vec{z}_i, \texttt{value\_decode}(\vec{H}_i))$$

### 4.3 OUTPUT VARIANCE TEST AND DECODING

Our approach incorporates the copula-based decoder to construct the joint distribution of the missing data from the observed latent vectors. By incorporating midpoint inference and local attention mechanisms, the decoder efficiently captures dependencies among neighboring imputed samples. However, it's worth noting that the system is vulnerable to errors, which can potentially impede the training process. To mitigate the error propagation, we implement an output variance test for each imputed data point. Specifically, for every imputation, we perform 10 predictions by sampling from the obtained joint distribution of the missing data. We then compare the output variance obtained from these predicted samples with a threshold, which is set to match the input data variance. If the output variance exceeds four times the threshold, we flag this predicted sample for exclusion in future imputations. In simpler terms, we mask out this predicted data point to prevent it from influencing future imputation processes. With a fixed window size, the decoder's complexity can be described as $O(nN)$, where $n$ is the number of time series variables. In summary, integrating the perceiver-based architecture with the tailored inference mechanisms significantly elevates the performance of PrACTiS, resulting in notable improvements in both efficiency and scalability.

## 5 EXPERIMENTAL STUDY

We present comprehensive experiments to showcase the computational efficiency of PrACTiS. First, we conduct memory consumption scaling experiments using synthetic random walk data to demonstrate the memory efficiency of our proposed model. Next, we evaluate the predictive capabilities of our model, comparing it against the state-of-the-art approaches, such as TACTiS (Drouin et al., 2022), GPVar (Salinas et al., 2019), SSAE-LSTM (Zhu et al., 2021), and deep autoregressive AR (Kalliovirta et al., 2015). Our evaluation spans across three unimodal time series

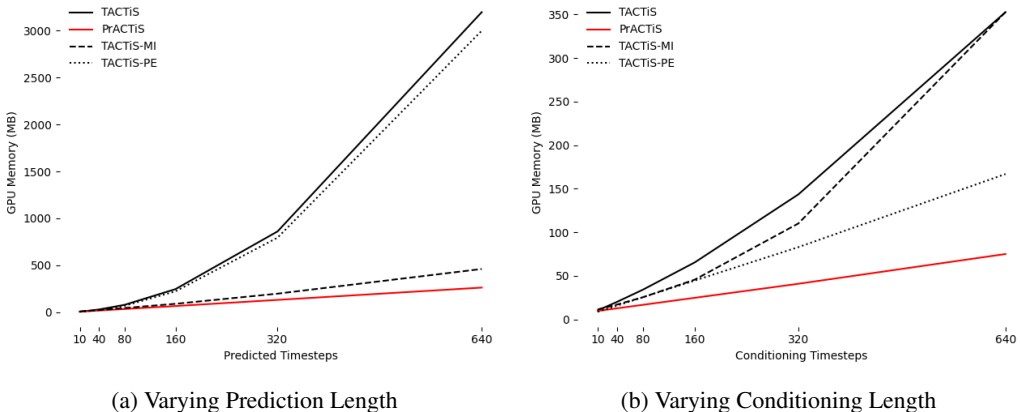

(a) Varying Prediction Length  (b) Varying Conditioning Length

Figure 3: Comparison of memory consumption of PrACTiS, TACTiS, TACTiS with perceiver-based encoder (TACTiS-PE), and TACTiS with midpoint imputation (TACTiS-MI) on a synthetic dataset with (a) varying prediction length and (b) varying conditioning length.

datasets from the Monash Time Series Forecasting Repository (Godahewa et al., 2021), including `electricity`, `traffic`, and `fred-md`, for short-term and long-term prediction tasks. Moreover, we evaluate the multi-modality capabilities of our perceiver-based model in three multimodal time series datasets from the UCI Machine Learning Repository (Dua & Graff, 2017), including `room occupation` (Candanedo, 2016), `interstate traffic` (Hogue, 2019), and `air quality` (Chen, 2019) datasets. The experimental results show the efficacy of our proposed PrACTiS model over other approaches in prediction performance and memory utilization.

## 5.1 MEMORY CONSUMPTION SCALING ON SYNTHETIC DATASETS

In this experiment, we evaluate the computational costs associated with our proposed PrACTiS model and the state-of-the-art TACTiS approach with respect to the quantity of observed and inferred data. Here, we use the synthetic random walk data with a synchronously-measured time series consisting of 10 variables, 10 observed time-steps, and 10 to-be-inferred time-steps. Additionally, we vary the number of observed and inferred time-steps to assess their impact. Our analysis extends to comparing our PrACTiS model with a perceiver-based variant of TACTiS (TACTiS-PE). It employs similar encoding and decoding mechanisms as TACTiS but leverages the perceiver-based encoder. We also consider TACTiS with a midpoint inference mechanism (TACTiS-MI). This model deduces data points using midpoint imputation and temporally local attention.

A comprehensive comparison of memory usage among these models when applied to a single input series is illustrated in Figure 3. Firstly, it shows the quadratic relationship between the computational cost of TACTiS and the quantity of input data. Secondly, it underscores the remarkable efficiency of the proposed PrACTiS in terms of memory utilization. Additionally, it showcases the improvements achieved by TACTiS variants over the original model. TACTiS-PE, which utilizes TACTiS' decoder, operates quadratically when dealing with inferred variables, thereby maintaining its quadratic scaling with respect to the number of predicted time-steps. Conversely, TACTiS-MI employs TACTiS' encoder, preserving its quadratic scaling with respect to the number of observed time steps. Overall, these results underscore the success of PrACTiS and the proposed inference mechanisms in efficiently mitigating the inherent quadratic scaling issue within TACTiS.

## 5.2 FORECASTING ON UNIMODAL DATASETS

In these experiments, we evaluate our proposed model's computational cost and inference performance across three real-world unimodal datasets. To begin, we employ the `fred-md` time series dataset, consisting of 20 input variables, each comprising 24 observed samples, with the goal of predicting 24 time-steps into the future. Appendix C provides insights into the models and their training procedures. Table 1 presents a comparative analysis of performance metrics for PrACTiS, TACTiS, GPVar, and AR models. We evaluate these models based on negative log-likelihoods (NLL), root-

Table 1: Comparison of memory usage and prediction performance between PrACTiS and other approaches in unimodal time series datasets, such as fred-md, traffic, and electricity.

| Approach | Params | Memory | Batches/s | NLL | RMSE-CM | CRPS |
|---|---|---|---|---|---|---|
| **fred-md – 24 timesteps prediction** | | | | | | |
| AR(24) | 6K | 3.7 MB | 37.4 | – | $7.0\pm_{0.5}E+2$ | $1.10\pm_{0.05}$ |
| GPVar | 78K | 1.39 GB | 11.7 | $42.3\pm_{0.6}$ | $6.8\pm_{0.5}E+2$ | $0.86\pm_{0.06}$ |
| TACTiS | 91K | 1.51 GB | 12.3 | $42.3\pm_{0.4}$ | $6.1\pm_{0.4}E+2$ | $0.74\pm_{0.05}$ |
| PrACTiS | 122K | 1.66 GB | 16.6 | $34.2\pm_{0.3}$ | $\mathbf{6.0\pm_{0.4}E+2}$ | $\mathbf{0.71\pm_{0.06}}$ |
| **traffic – 48 timesteps prediction** | | | | | | |
| AR(48) | 20K | 11.5 MB | 10.31 | – | $0.053\pm_{0.005}$ | $0.431\pm_{0.004}$ |
| GPVar | 78K | 4.67 GB | 5.81 | $204.6\pm_{0.8}$ | $0.044\pm_{0.003}$ | $0.215\pm_{0.008}$ |
| TACTiS | 91K | 5.52 GB | 5.84 | $198.7\pm_{0.6}$ | $0.035\pm_{0.002}$ | $0.181\pm_{0.009}$ |
| PrACTiS | 122K | 2.75 GB | 5.95 | $188.7\pm_{0.6}$ | $\mathbf{0.028\pm_{0.002}}$ | $\mathbf{0.162\pm_{0.006}}$ |
| **electricity – 48 timesteps prediction** | | | | | | |
| AR(48) | 20K | 11.6 MB | 10.34 | – | $90\pm_{0.1}$ | $0.149\pm_{0.001}$ |
| GPVar | 78K | 4.78 GB | 5.76 | $185.6\pm_{0.5}$ | $62\pm_{0.1}$ | $0.060\pm_{0.001}$ |
| TACTiS | 91K | 5.42 GB | 5.81 | $182.3\pm_{0.6}$ | $49\pm_{0.1}$ | $0.060\pm_{0.001}$ |
| PrACTiS | 122K | 2.73 GB | 5.93 | $177.8\pm_{0.8}$ | $\mathbf{42\pm_{0.1}}$ | $\mathbf{0.056\pm_{0.001}}$ |
| **electricity – 672 timesteps prediction** | | | | | | |
| AR(672) | 270K | 47.7 MB | 1.74 | – | $159\pm_{0.8}$ | $0.290\pm_{0.02}$ |
| GPVar | 78K | 4.81 GB | 3.48 | $3.5\pm_{0.4}E+3$ | $147\pm_{0.5}$ | $0.198\pm_{0.005}$ |
| TACTiS | 91K | 4.81 GB | 3.65 | $2.8\pm_{0.2}E+3$ | $141\pm_{0.3}$ | $0.186\pm_{0.006}$ |
| PrACTiS | 122K | 372 MB | 18.3 | $185\pm_{0.9}$ | $\mathbf{98\pm_{0.1}}$ | $\mathbf{0.133\pm_{0.001}}$ |

mean-squared-errors of conditional expectations (RMSE-CM), and continuous ranked probability scores (CRPS). In Figures 4, 5, and 6, we demonstrate example inferences generated by PrACTiS, TACTiS, and AR models, respectively. Our proposed model outperforms GPVar and AR while achieving competitive results with TACTiS in both RMSE-CM and CRPS metrics.

Next, we utilize `traffic` time series data with 20 input variables, each with 48 observed samples to predict 48 time-steps ahead. The performance comparison between PrACTiS, TACTiS, GPVar, AR is demonstrated in Table 1. We also present example inferences from PrACTiS, TACTiS, and AR in Figures 7, 8, and 9 respectively. Here, our proposed model demonstrates a significant performance advantage over TACTiS, GPVar, and AR, excelling in both RMSE-CM and CRPS metrics. Notably, we achieve 20% improvement over TACTiS in terms of RMSE-CM. Furthermore, the reported numbers of parameters and memory usage highlight the efficiency of PrACTiS, which utilizes less than 50% of the memory compared to TACTiS and GPVar.

Finally, we evaluate PrACTiS in the context of short-term and long-term prediction tasks using the `electricity` dataset. In the short-term prediction experiment, we utilize 20 variables, each spanning 48 observed time-steps, to forecast 48 time-steps into the future. As shown in Table 1, our proposed model significantly outperforms other approaches, boasting a 14% improvement in RMSE-CM compared to TACTiS, all while utilizing just 50% of available memory. Figures 10, 11, and 12 illustrate the examples of predictions made by PrACTiS, TACTiS, and AR(48), respectively. For the long-term prediction task, we work with 10 variables, each encompassing 672 observed time-steps, to predict the subsequent 672 time-steps. This experiment provides valuable insights into the capabilities of these models on a large-scale dataset. Visual representations of the predictions from PrACTiS, TACTiS, and AR are shown in Figures 13, 14, and 15, respectively. In this scenario, PrACTiS demonstrates a significant performance advantage over TACTiS, excelling in both RMSE-CM and CRPS while utilizing only 10% of available memory. It's noteworthy that PrACTiS manages to capture the seasonal patterns in the data, albeit not as accurately as in the short-term task. Conversely, TACTiS and other methods face inherent challenges when dealing with extended time series. In particular, TACTiS struggles to model the underlying seasonal structures within the data, resulting in less reliable performance when tasked with long-term predictions.

Table 2: Comparison of memory usage and prediction performance between PrACTiS and other approaches in multimodal time series datasets, such as room occupation, interstate traffic, air quality.

| Approach | Params | Memory | RMSE-CM | Use/No Use | High/Low $CO_2$ |
|---|---|---|---|---|---|
| **room occupation – 6 features attributions** | | | | | |
| SSAE-LSTM | 76K | 5.22 GB | $0.056 \pm 0.002$ | 97.1% | 96.5% |
| TACTiS | 91K | 6.38 GB | $0.031 \pm 0.001$ | 98.1% | 97.7% |
| PrACTiS | 122K | 3.09 GB | $\mathbf{0.018 \pm 0.001}$ | **98.9%** | **98.4%** |

| Approach | Params | Memory | RMSE-CM | Rain/No Rain | High/Low Traffic |
|---|---|---|---|---|---|
| **interstate traffic – 8 features attributions** | | | | | |
| SSAE-LSTM | 76K | 5.68 GB | $0.083 \pm 0.004$ | 95.3% | 94.6% |
| TACTiS | 91K | 7.13 GB | $0.065 \pm 0.003$ | 96.7% | 96.1% |
| PrACTiS | 122K | 3.22 GB | $\mathbf{0.027 \pm 0.003}$ | **98.2%** | **97.8%** |

| Approach | Params | Memory | RMSE-CM | Rain/No Rain | High/Low PM2.5 |
|---|---|---|---|---|---|
| **air quality – 12 features attributions** | | | | | |
| SSAE-LSTM | 76K | 6.17 GB | $0.106 \pm 0.006$ | 93.7% | 93.4% |
| TACTiS | 91K | 8.83 GB | $0.074 \pm 0.005$ | 95.8% | 94.9% |
| PrACTiS | 122K | 3.41 GB | $\mathbf{0.022 \pm 0.004}$ | **98.5%** | **98.1%** |

## 5.3 FORECASTING ON MULTIMODAL DATASETS

In the multimodal experiments, we first evaluate the predictive capabilities of PrACTiS on the `room occupation` dataset (Candanedo, 2016). This dataset is multimodal, consisting of 6 feature attributes related to room conditions, such as temperature, humidity, and $CO_2$ levels. A detailed dataset description is available in Appendix D. Here, we conduct a comparative analysis with TAC-TiS (Drouin et al., 2022) and SSAE-LSTM (Zhu et al., 2021). Both of these methods employ a strategy of concatenating all feature attributes at each time-step for prediction. The performance results, as presented in Table 2, consist of measures such as average RMSE-CM, room occupation detection accuracy, and high $CO_2$ detection accuracy. The memory usage is also provided to highlight the efficiency of PrACTiS when achieving 40% reduction in RMSE-CM compared to TACTiS, while utilizing only half of the memory resources. We then extend our experimentation to the `interstate traffic` dataset (Hogue, 2019). This dataset comprises 8 feature attributes related to weather conditions (e.g., temperature, snow), holiday status, and traffic volume. Table 2 illustrates that PrACTiS significantly outperforms other approaches while maintaining linear memory usage. Notably, our approach achieves a 58% improvement in RMSE-CM compared to TACTiS and consistently excels in prediction tasks related to detecting rain and high traffic. Finally, we evaluate the performance of our approach on the `air quality` dataset (Chen, 2019), which encompasses 12 variables, each with 12 feature attributes, including 6 pollution-related features (e.g., PM2.5, PM10) and 6 weather-related features (e.g., temperature, rain). Table 2 showcases the performance comparison between PrACTiS and other approaches, with our model achieving a remarkable 70% improvement in RMSE-CM compared to TACTiS while utilizing only 40% of the memory resources.

The results of our experiments illustrate that PrACTiS exhibits increasing efficiency as both the prediction length and the number of feature attributes grow in scale. This efficiency gain becomes particularly pronounced when compared to TACTiS and other existing approaches. In essence, the perceiver-based architecture and the midpoint inference mechanism have proven to be highly effective in addressing the challenges posed by complex multimodal datasets.

## 6 CONCLUSIONS

We introduce a new approach to time series forecasting, harnessing the power of cross-attention with a copula-based structure. Our model excels at encoding the global description of partially-observed time series into latent representations, effectively reducing computational complexity. It also integrates a temporally local attention mechanism through midpoint inference, which restricts token attention to those with the utmost temporal relevance to their conditioning for precise conditional modeling. Our experiments underscore the substantial reduction in computational costs when modeling extensive time series data, while delivering competitive inference performance compared with the state-of-the-art methods.

## REPRODUCIBILITY STATEMENT

We provide a comprehensive breakdown of the experimental setup details for the memory consumption experiment in Appendix B, encompassing the dataset description and the model parameters. Additional details on the unimodal data experiment are shown in Appendix C, including the detailed unimodal datasets descriptions and the complete experimental setup (i.e., the training parameters for the PrACTiS, TACTiS models). The examples of the predicted samples using these models are illustrated in Appendix E. For the multimodal data experiment, we provide a thorough description of multimodal datasets and the experimental training procedure in Appendix D.

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

## A ADVANTAGES AND DISADVANTAGES OF PRACTIS

Throughout extensive experiments on unimodal and multimodal datasets, we have observed that PrACTiS effectively learns to model time series data and contributes to improved prediction performance. Nevertheless, it is worth noting that in certain instances, the utilization of PrACTiS may result in a significant increase in memory consumption without proportional gains in performance. For instance, for shorter time series, such as the `fred-md` dataset in the Monash Time Series Forecasting Repository (Godahewa et al., 2021), PrACTiS may require increased time and computational resources for training compared to the TACTiS approach. This heightened resource requirement stems from the overhead associated with the cross-attention mechanism, which is designed to map input embeddings into latent embeddings. In this experiment, PrACTiS obtains insignificant improvement over TACTiS, but utilizes more memory resources. In contrast, when dealing with long and complex time series, such as those present in the Monash Time Series Forecasting Repository (Godahewa et al., 2021) (e.g., `traffic`, `electricity` datasets), and the UCI Machine Learning Repository (Dua & Graff, 2017) (`room occupation` dataset (Candanedo, 2016), `interstate traffic` (Hogue, 2019), and `air quality` (Chen, 2019) datasets), our experiments clearly demonstrate the advantages of PrACTiS's enhanced scalability, as exemplified by the outcomes of our memory scaling experiments. These advancements highlight PrACTiS models' ability to train efficiently on hardware configurations with more modest computational resource allocations, rendering them highly suitable for a wide range of time series modeling tasks. Furthermore, PrACTiS exhibits the capacity to handle more complex tasks with a linear increase in memory usage, allowing for greater flexibility in model design. In summary, PrACTiS simplifies the modeling of time series data that would otherwise pose computational challenges when employing TACTiS or alternative

methodologies. As a result, PrACTiS can adeptly learn to model time series data while consuming only a fraction of the memory resources, all while achieving highly competitive performance levels.

Overall, PrACTiS shows potential applications in the generative models for handling missing data. Its utilization of latent variables to deduce a global state and enhance computational efficiency bears a striking resemblance to how other generative models leverage their latent variables for efficient missing data inference. The fusion of this latent space with an attentional generative mechanism empowers PrACTiS to overcome many of the inherent architectural challenges encountered when applying existing generative models to missing data tasks. Attentional models, which replace complex architectural details with information encoded within their input data, may prove to be ideally suited for addressing missing data challenges.

## B  ADDITIONAL DETAILS FOR MEMORY CONSUMPTION EXPERIMENTS

To illustrate how the computational demands of PrACTiS and TACTiS models scale in relation to the quantity of observed and inferred data, we investigate the memory usage of these two architectures during training using synthetic random walk data. In this experimental setup, we initiate with a synchronously-measured Random Walk time series comprising 10 variables, encompassing 10 observed time-steps and an additional 10 time-steps to be inferred. We then systematically vary both the number of observed time-steps and the number of inferred time-steps. The model parameters employed for our memory scaling experiments with TACTiS and PrACTiS are detailed in Table 3.

Our analysis also extends to comparing our PrACTiS model with a perceiver-based variant of TACTiS (TACTiS-PE). It employs similar encoding and decoding mechanisms as TACTiS but leverages the perceiver-based encoder. We also consider TACTiS with a midpoint inference mechanism (TACTiS-MI). This model deduces data points using midpoint imputation and temporally local attention. As shown in Figure 3, both of these models always outperform the original TACTiS, demonstrating the efficacy of the added mechanism.

## C  ADDITIONAL DETAILS FOR INFERENCE PERFORMANCE EXPERIMENTS

The TACTiS model parameters employed for these experiments were adopted from the configuration used by Drouin et al. (2022). We also adopt these parameters as the foundation for establishing a comparable PrACTiS model. Additionally, we employ deep AR(d) models, characterized as feed-forward models that take the previous $d$ time-steps as inputs to predict the current time-step. Below, in Table 4, we provide a comprehensive listing of the model parameters utilized for our deep AR, TACTiS, and PrACTiS models.

### C.1  FURTHER DETAILS FOR SHORT FORECASTING

The datasets, namely `fred-md`, `traffic`, and `electricity`, all consist of 20 input variables. Our model training process for these datasets follows a consistent protocol: we use batch sizes of 24 for the `fred-md` dataset and 48 for the `traffic` and `electricity` datasets. This training routine extends over 100 epochs, each comprising 512 batches. For optimizing all our models, we employ the RMSProp optimizer (Hinton et al., 2012) with an initial learning rate set at $1e - 3$.

### C.2  FURTHER DETAILS FOR LONG FORECASTING

Given the considerable computational demands associated with forecasting 672 time-steps in this experiment, we made the strategic decision to reduce the batch size to 1 in order to facilitate the training of the TACTiS model. However, this adjustment presented challenges when applying PrACTiS's midpoint inference scheme, as its initial forecasts extended across hundreds of time steps, leading to suboptimal performance. To overcome this limitation and achieve substantially improved results in the realm of long-term forecasting, we refined the midpoint inference scheme. This refinement involved introducing greater autoregressive behavior at the outset, with a carefully designed sampling order that ensured each time-step remained within a user-defined maximum interval from the nearest conditioning variable. In this experiment, we set a relatively aggressive maximum interval of three time-steps, and the subsequent results reflect the impact of this adjustment.

Table 3: Model Parameters for Memory Consumption Scaling Experiment.

(a) TACTiS

| Input Encoding | |
|---|---|
| Series Embedding Dim. | 5 |
| Input Encoder Layers | 3 |
| Positional Encoding | |
| Dropout | 0.0 |
| Temporal Encoder | |
| Attention Layers | 3 |
| Attention Heads | 3 |
| Attention Dim. | 16 |
| Attention Feedforward Dim. | 16 |
| Dropout | 0.0 |
| Copula Decoder | |
| Min. u | 0.01 |
| Max. u | 0.99 |
| Attentional Copula | |
| Attention Layers | 3 |
| Attention Heads | 3 |
| Attention Dim. | 16 |
| Feedforward Dim. | 16 |
| Feedforward Layers | 3 |
| Resolution | 50 |
| Marginal Flow | |
| Feedforward Layers | 2 |
| Feedforward Dim. | 8 |
| Flow Layers | 2 |
| Flow Dim. | 8 |

(b) PrACTiS

| Input Encoding | |
|---|---|
| Series Embedding Dim. | 5 |
| Input Encoder Layers | 3 |
| Positional Encoding | |
| Dropout | 0.0 |
| Perceiver Encoder | |
| Num. Latents | 64 |
| Latent Dim. | 48 |
| Attention Layers | 3 |
| Self-attention Heads | 3 |
| Cross-attention Heads | 3 |
| Dropout | 0.0 |
| Perceiver Decoder | |
| Cross-attention Heads | 3 |
| Copula Decoder | |
| Min. u | 0.01 |
| Max. u | 0.99 |
| Attentional Copula | |
| Attention Layers | 3 |
| Attention Heads | 3 |
| Attention Dim. | 16 |
| Feedforward Dim. | 16 |
| Feedforward Layers | 3 |
| Resolution | 50 |
| Marginal Flow | |
| Feedforward Layers | 2 |
| Feedforward Dim. | 8 |
| Flow Layers | 2 |
| Flow Dim. | 8 |

It's important to highlight that the autoregressive refinement made in the midpoint inference order doe not alter the computational complexity of training a PrACTiS model. While this adjustment has the potential to improve forecasting accuracy in specific situations, it's worth acknowledging that it can also pose challenges when predicting time-steps that fall between observed data points.

## D  ADDITIONAL DETAILS FOR MULTIMODAL EXPERIMENTS

The `room occupation` dataset (Candanedo, 2016) is a comprehensive multimodal dataset encompassing six distinct feature attributes that capture room conditions and room occupancy status (i.e., the primary output). These attributes include temperature, relative humidity, humidity ratio, light levels, and $CO_2$ concentrations. In this experiment, our model is trained to forecast 48 time-steps ahead, utilizing historical data spanning the preceding 48 time-steps. The evaluation of predictive performance is based on the average RMSE-CM across all six attributes. Furthermore, we undertake two classification tasks: the first task involves predicting room occupancy, while the second task focuses on detecting high $CO_2$ levels (i.e., levels exceeding 700 ppm).

The `interstate traffic` dataset (Hogue, 2019) presents a collection of multimodal traffic data samples, encompassing eight feature attributes that capture a wide range of information. These attributes are associated with diverse aspects, including weather conditions, temporal factors, holiday status, and traffic volume (i.e., primary output). Here, the weather-related attributes include temperature, precipitation (rain and snow), cloud cover, and the categorization of weather conditions. In this experimental setup, our model is rigorously trained to predict traffic conditions up to 48 time-steps into the future, leveraging historical data spanning the preceding 48 time-steps. To assess predictive performance, we utilize RMSE-CM calculated across all eight attributes. Addition-

Table 4: Model Parameters for Performance Experiments.

(a) Deep AR(d)

| Num. Layers | 3 |
|---|---|
| Hidden Dim. | d |

(b) TACTiS

| Input Encoding | |
|---|---|
| Series Embedding Dim. | 5 |
| Input Encoder Layers | 3 |
| Positional Encoding | |
| Dropout | 0.01 |
| Temporal Encoder | |
| Attention Layers | 2 |
| Attention Heads | 2 |
| Attention Dim. | 24 |
| Attention Feedforward Dim. | 24 |
| Dropout | 0.01 |
| Copula Decoder | |
| Min. u | 0.05 |
| Max. u | 0.95 |
| Attentional Copula | |
| Attention Layers | 1 |
| Attention Heads | 3 |
| Attention Dim. | 16 |
| Feedforward Dim. | 48 |
| Feedforward Layers | 1 |
| Resolution | 20 |
| Marginal Flow | |
| Feedforward Layers | 1 |
| Feedforward Dim. | 48 |
| Flow Layers | 3 |
| Flow Dim. | 16 |

(c) PrACTiS

| Input Encoding | |
|---|---|
| Series Embedding Dim. | 5 |
| Input Encoder Layers | 3 |
| Positional Encoding | |
| Dropout | 0.01 |
| Perceiver Encoder | |
| Num. Latents | 256 |
| Latent Dim. | 48 |
| Attention Layers | 2 |
| Self-attention Heads | 3 |
| Cross-attention Heads | 3 |
| Dropout | 0.01 |
| Perceiver Decoder | |
| Cross-attention Heads | 3 |
| Copula Decoder | |
| Min. u | 0.05 |
| Max. u | 0.95 |
| Attentional Copula | |
| Attention Layers | 1 |
| Attention Heads | 3 |
| Attention Dim. | 16 |
| Feedforward Dim. | 48 |
| Feedforward Layers | 1 |
| Resolution | 20 |
| Marginal Flow | |
| Feedforward Layers | 1 |
| Feedforward Dim. | 48 |
| Flow Layers | 3 |
| Flow Dim. | 16 |

ally, we investigate two classification tasks: firstly, identifying instances of rainy weather conditions, and secondly, detecting periods of high traffic volume (i.e., volumes exceeding 2000 cars).

The `air quality` (Chen, 2019) dataset comprises pollution measures and weather-related metrics data. It encompasses 12 variables, each with 12 feature attributes, including 6 pollution-related features and 6 weather-related features. The pollution-related features include PM2.5, PM10, $SO_2$, $NO_2$, CO, $O_3$ concentrations. The weather-related features consist of temperature, dew point temperature, pressure, precipitation, wind direction and speed. Similarly, our model is trained to forecast 48 time-steps into the future, leveraging historical data spanning the preceding 48 time-steps. To assess the quality of our predictions, we employ the average RMSE-CM calculated across all attributes. Moreover, we tackle two classification tasks: firstly, identifying instances of rainy weather conditions, and secondly, detecting periods with elevated PM2.5 levels, specifically those exceeding 80 $\mu g/m^3$. The detailed results of these experiment can be found in Table 2.

# E   FORCASTING SAMPLES

Here, we provide some predicted samples produced using the PrACTiS, TACTiS, and AR models. First, Figure 4, 5, and 6 illustrate the predicted samples from the short-term task (i.e., 24 time-steps, which corresponds to 2 years) in `fred-md` dataset using PrACTiS, TACTiS, and AR models, respectively. Next, Figure 7, 8, and 9 demonstrate the short-term predicted samples (i.e., 48

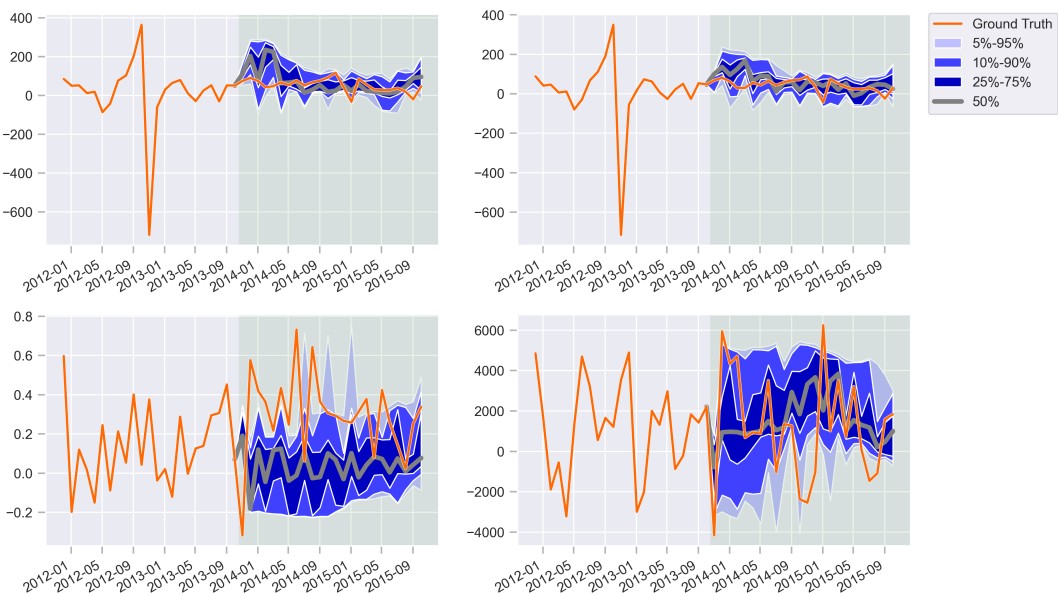

Figure 4: The predicted samples by the PrACTiS model for two-year forecasts, corresponding to 24 time-steps, conditioned on two-year historical data in `fred-md` dataset.

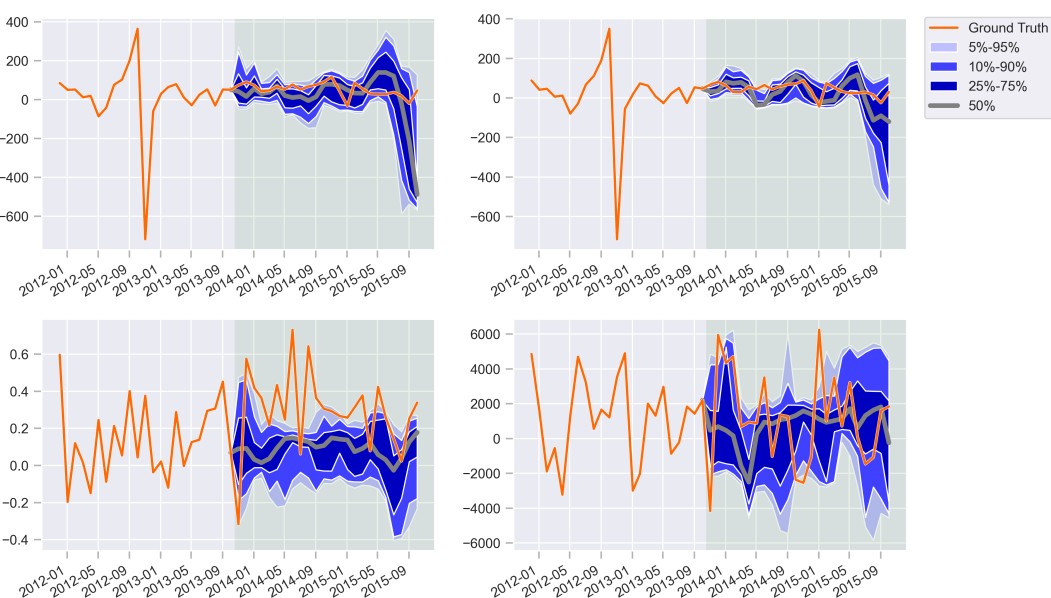

Figure 5: The predicted samples by the TACTiS model for two-year forecasts, corresponding to 24 time-steps, conditioned on two-year historical data in `fred-md` dataset.

time-step, corresponding to 2 days) in `traffic` dataset using PrACTiS, TACTiS, and AR models, respectively. Similarly, the short-term predicted samples (i.e., 48 time-step, corresponding to 2 days) in `electricity` dataset using PrACTiS, TACTiS, and AR models are shown in Figure 10, 11, and 12, respectively. Lastly, Figure 13, 14, and 15 illustrate the last 4-day predicted samples from the long-term task (i.e., 672 time-step, corresponding to 1 month) in `electricity` dataset using PrACTiS, TACTiS, and AR models, respectively.

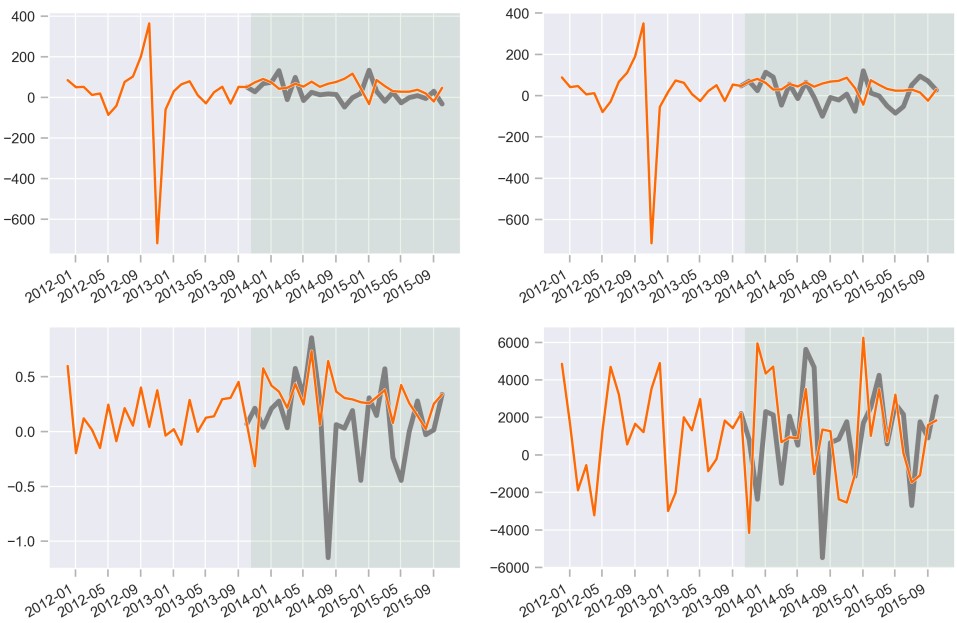

Figure 6: The predicted samples by the AR(24) model for two-year forecasts, corresponding to 24 time-steps, conditioned on two-year historical data in `fred-md` dataset.

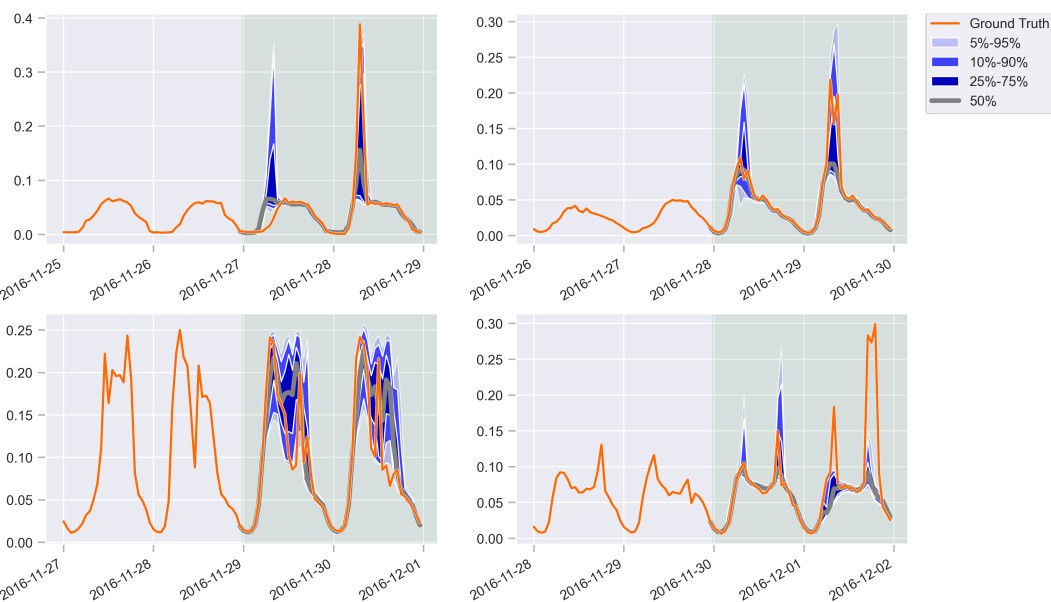

Figure 7: The predicted samples by the PrACTiS model for two-day forecasts, corresponding to 48 time-steps, conditioned on two-day historical data in `traffic` dataset.

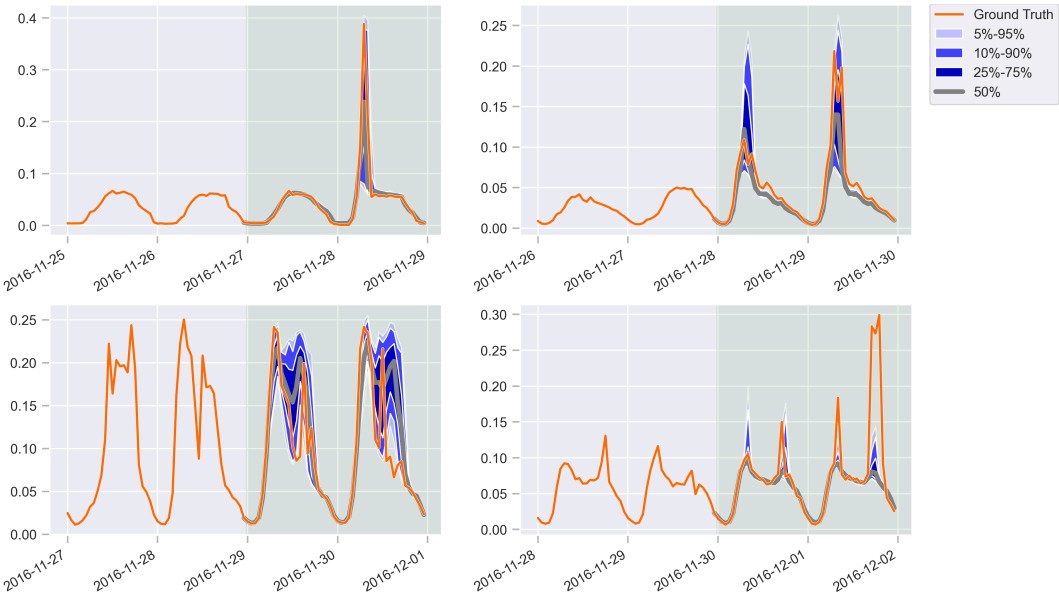

Figure 8: The predicted samples by the TACTiS model for two-day forecasts, corresponding to 48 time-steps, conditioned on two-day historical data in `traffic` dataset.

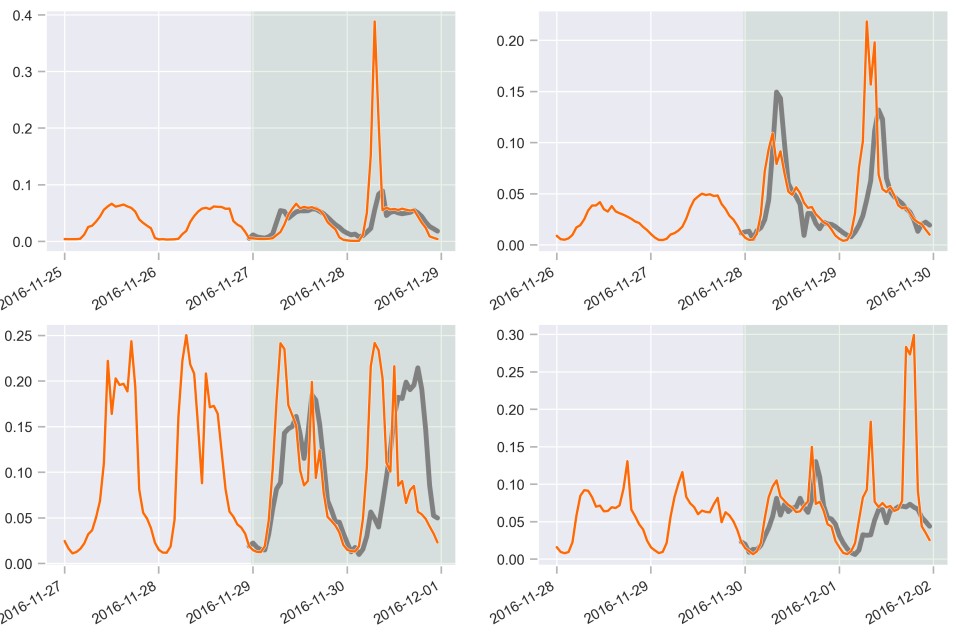

Figure 9: The predicted samples by the AR(48) model for two-day forecasts, corresponding to 48 time-steps, conditioned on two-day historical data in `traffic` dataset.

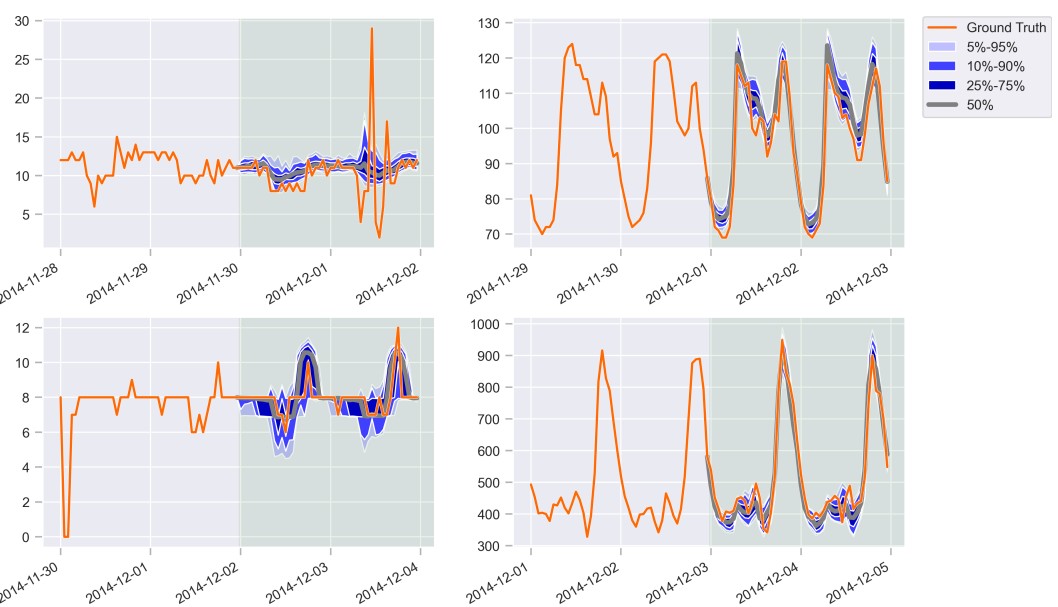

Figure 10: The predicted samples by the PrACTiS model for two-day forecasts, corresponding to 48 time-steps, conditioned on two-day historical data in `electricity` dataset.

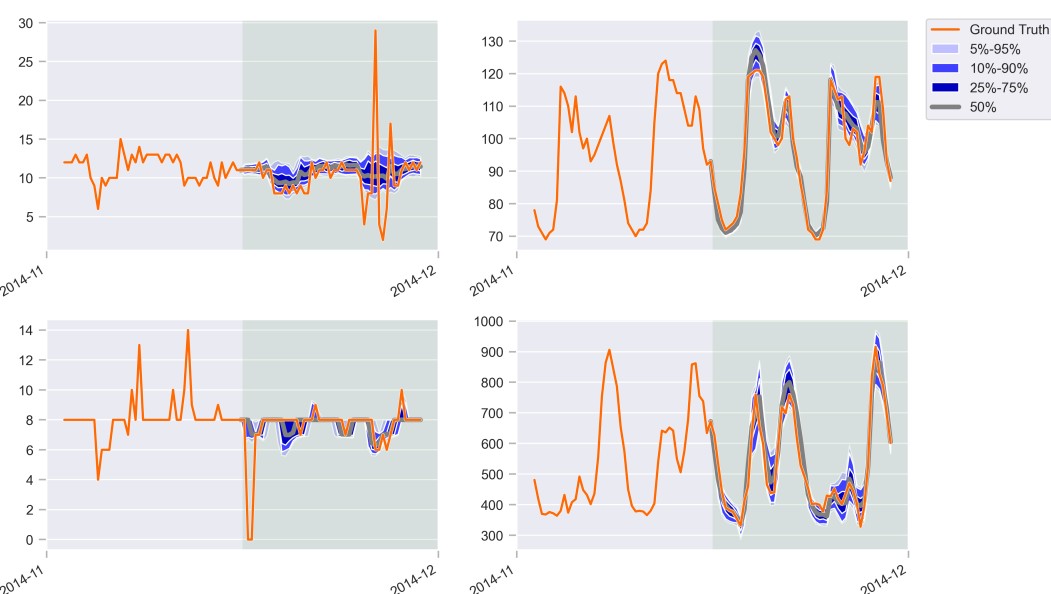

Figure 11: The predicted samples by the TACTiS model for two-day forecasts, corresponding to 48 time-steps, conditioned on two-day historical data in `electricity` dataset.

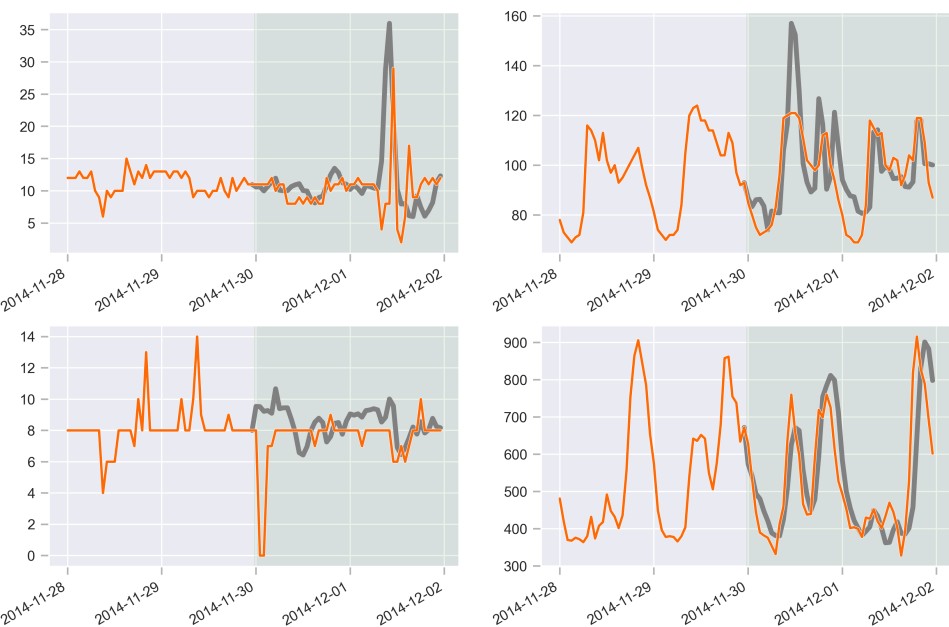

Figure 12: The predicted samples by the AR(48) model for two-day forecasts, corresponding to 48 time-steps, conditioned on two-day historical data in `electricity` dataset.

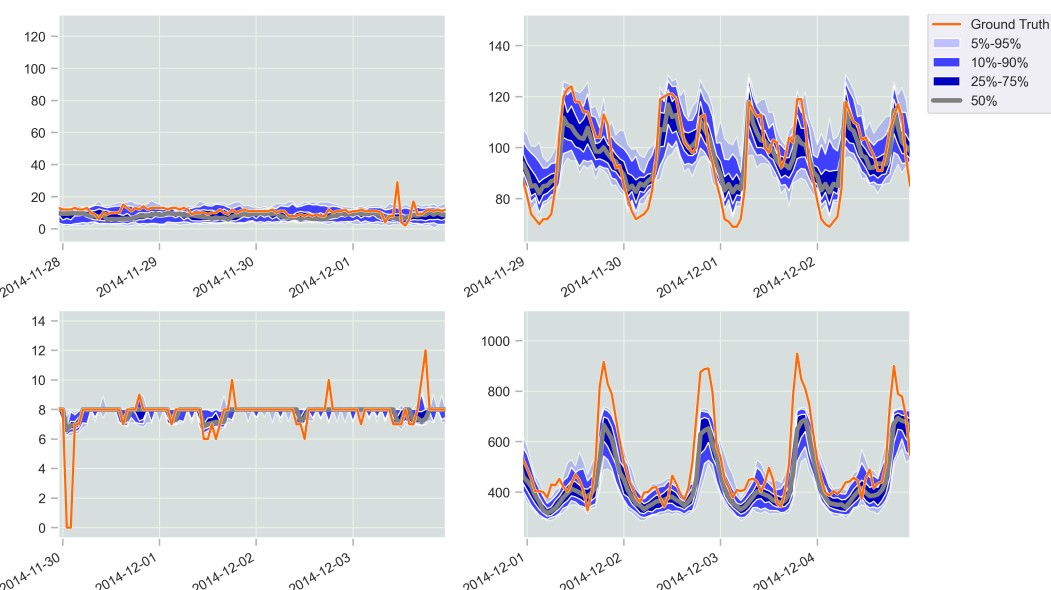

Figure 13: The final four-day predicted samples by the PrACTiS model for one-month forecasts, corresponding to 672 time-steps, conditioned on one-month historical data in `electricity` dataset.

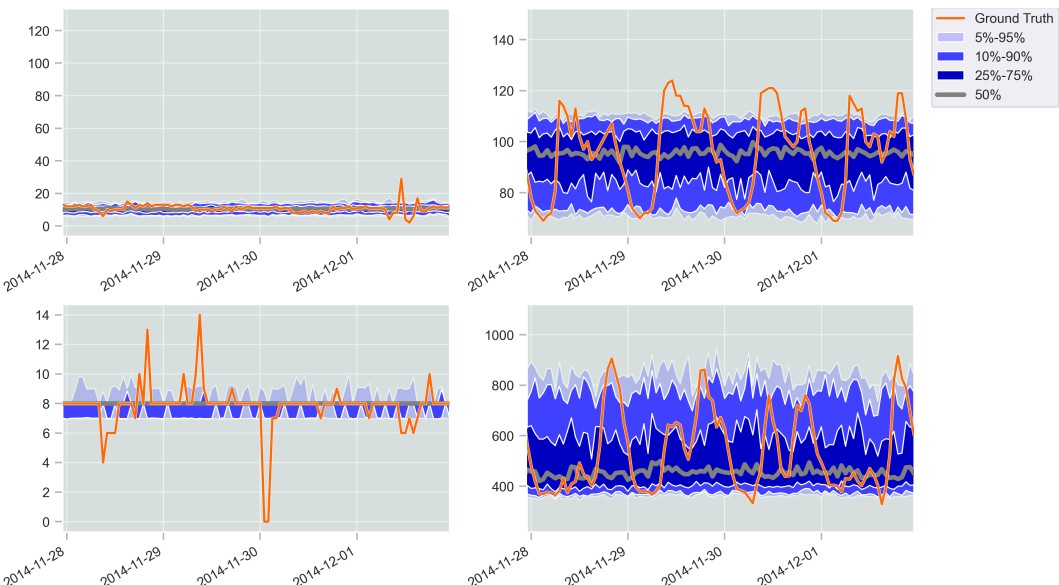

Figure 14: The final four-day predicted samples by the TACTiS model for one-month forecasts, corresponding to 672 time-steps, conditioned on one-month historical data in `electricity` dataset.

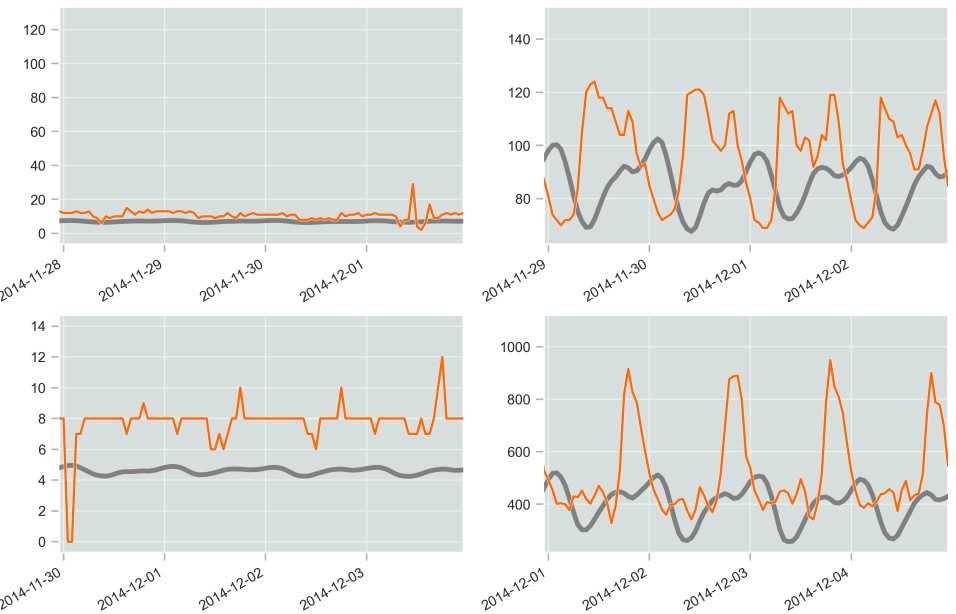

Figure 15: The final four-day predicted samples by the AR(672) model for one-month forecasts, corresponding to 672 time-steps, conditioned on one-month historical data in `electricity` dataset.

