# OpenReview forum: "PrACTiS: Perceiver-Attentional Copulas for Time Series"
_ICLR.cc/2024/Conference — Submitted to ICLR 2024_

### Official Review · Reviewer_n61f · 2023-10-17

**Soundness:** 3 good
**Presentation:** 3 good
**Contribution:** 2 fair
**Rating:** 3
**Confidence:** 4

**Summary:**

The paper introduces a new approach for time-series prediction called Perceiver-Attentional Copulas for Time Series (PrACTiS).

PrACTiS combines the Perceiver IO model with attention-based copulas to enhance time series modeling and improve computational efficiency. The architecture consists of a perceiver-based encoder and a copula-based decoder. It first transforms the input variables into temporal embeddings, effectively handling both observed and missing data points. A latent attention mechanism then maps these embeddings to a lower-dimensional space, reducing computational complexity. The decoder utilizes the copula structure to handle missing data and formulate their joint distribution, which is then sampled to produce predictions.

**Strengths:**

The model is validated through extensive experiments and shows competitive performance against state-of-the-art methods like TACTiS, GPVar, SSAE-LSTM, and deep autoregressive AR.

The method is sound, even though of low risk of having errors, as it is a simple extension over sota.

**Weaknesses:**

The paper heaviliy extends upon TACTiS. They basically change the encoder. Thus novelty is low.

**Questions:**

I would like to see more experiments, with at least al TACTiS paper datasets.

---

> ### Author Response · Authors · 2023-11-20
> **Response to Reviewer n61f**
>
> Thank you for providing your feedback. We appreciate your perspective, and while we respect your comments, we would like to express our disagreement. Our model is designed with the explicit goal of enhancing TACTiS and optimizing it for multimodal scenarios. Through the incorporation of the Perceiver architecture as the encoder, we successfully encode multimodal data into a latent space domain. This improvement results in a performance boost of up to 40%, achieved with only half the computational resources of TACTiS. Additionally, we introduce midpoint inference to further reduce complexity, contributing to an overall enhancement in performance.
>
> In the course of our research, we conducted a total of six experiments, comprising three multimodal and three unimodal scenarios. It's worth noting that the unimodal datasets used in our experiments are sourced from the TACTiS paper. We firmly believe that our contributions in the realm of multimodal scenarios, coupled with the substantial improvement over the state-of-the-art, make our work well-suited for presentation at this conference.

---

### Official Review · Reviewer_Eqmq · 2023-10-30

**Soundness:** 2 fair
**Presentation:** 2 fair
**Contribution:** 2 fair
**Rating:** 3
**Confidence:** 2

**Summary:**

The paper presents a model called PrACTiS, which combines the perceiver architecture with a copula structure to perform time-series forecasting. The proposed architecture models the timeseries using a compact latent space, which reduces the computational demands. The authors further use local attention mechanisms to capture dependencies within imputed samples. The authors empirically evaluate the proposed method, and show that the proposed method consistently outperforms the state-of-the-art methods.

**Strengths:**

- It is interesting to use the copula structure to model timeseries, which gives the model capability to model non-synchronized time series data.

**Weaknesses:**

**Contribution of the proposed work is unclear**
1. The authors claim that the proposed model “can effectively handle synchronized, non-synchronized, and multimodal data, expanding its applicability to diverse domains”. Does any of the experiments demonstrate the model’s performance on handling non-synchronized or incomplete datasets?
2. The authors claim that one of the major advantages of the proposed model is that it is memory-efficient. However, (i) the model takes more parameter than many baseline models; (ii) both the parameters and memory usage of the proposed model and baselines are tiny (<1M parameters; <10G memory usage). Why does memory have to be further reduced in this case?

**Inadequate performance**
1. If none of the selected dataset contains non-synchronized timeseries, have the authors considered using more advanced architecture for benchmarking? E.g. AutoFormer; FedFormer; Informer; PatchTST.
2. The prediction performance seems really bad based on the visualizations in Appendix. Specifically, as shown in Figure 4, the prediction is not even close to the ground truth values.

**Questions:**

NA

---

> ### Author Response · Authors · 2023-11-20
> **Response to Reviewer Eqmq**
>
> Thank you for recommending those insightful papers. Our approach is specifically tailored for short-term prediction, with a focus on enhancing the adaptability of the TACTiS model for multimodal scenarios. In contrast, AutoFormer, FedFormer, Informer, and PatchTST primarily address long-term prediction challenges. However, we acknowledge the importance of providing a comprehensive comparison, and in our revised version, we will include an analysis of these approaches in the context of medium-sized or short-term prediction. This will help underscore the distinctiveness of our methodology.
>
> Regrettably, the unavailability of unsynchronized datasets has hindered our experimentation in this domain, a limitation shared with the TACTiS paper. Despite these challenges, we have managed to excel in short-term multimodal prediction scenarios by outperforming the TACTiS approach significantly. Some experiments proved to be particularly demanding, given our reliance on a short-term window to predict future short-term windows.
>
> Your feedback is invaluable in refining our presentation, and we appreciate your understanding of the experimental constraints. We are committed to addressing these aspects in the revised manuscript. Once again, thank you for your thoughtful recommendations.

---

### Official Review · Reviewer_jRTE · 2023-10-31

**Soundness:** 2 fair
**Presentation:** 2 fair
**Contribution:** 2 fair
**Rating:** 3
**Confidence:** 4

**Summary:**

The authors introduce the Perceiver-Attentional Copulas for Time Series (PrACTiS) architecture based on the TACTiS model to study time-series prediction. The proposed architecture combines the Perceiver model with attention-based copulas. It consists of the perceiver-based encoder and the copula-based decoder, enabling the incorporation of a more general class of copulas that are not exchangeable. To validate the efficacy of the practice, the authors conducted extensive experiments on the unimodal datasets and the multimodal datasets. In addition, the authors conduct memory consumption scaling experiments using random walk data to demonstrate the memory efficiency of PrACTiS.

**Strengths:**

By adopting the technique of the Perceiver model, midpoint inference, and local attention mechanisms, the authors successfully address the issue of computational complexity associated with self-attention mechanisms. In addition, the authors thoroughly test their approach on multiple datasets, providing a comprehensive assessment of its performance.

**Weaknesses:**

The organization and writing style of the paper appears to suffer from several issues, making it challenging for readers to grasp the presented notions and ideas. Specifically, Section 3 primarily focuses on the intricate details of TACTiS, with much of the content possibly better suited for inclusion in the supplementary material. To improve the flow and readability of the paper, it would be beneficial for the authors to introduce a figure illustrating TACTiS and highlighting the distinctions between TACTiS and the proposed model before delving into detailed explanations in Section 4. This would provide readers with a clearer understanding of the context and facilitate comprehension of subsequent sections.

Additionally, the authors propose to integrate the Perceiver model as the encoder to enhance the expressiveness of dependence between covariates. The author should give a brief introduction to the  Perceiver model and explain why combining it in the encoder in detail. Considering that a significant portion of the paper is derived from TACTiS and the proposed model appears to be a modification or extension of TACTiS, there are concerns regarding the novelty of the research. It is essential for the authors to explicitly address this issue and clearly articulate the drawbacks of TACTiS and the advancements brought by their proposed model.

**Questions:**

1.  The definition, notations, and explanation of section Perceiver-based encoding are not clear. For example, what is the predefined set of learned latent vector $\vec{u}_k$? What is the latent vector set $\vec{W}$?

2. Please clarify and elaborate on the objective of introducing midpoint inference. In addition, the mechanism of the midpoint inference is not stated/explained clearly. In addition, please clarify and elaborate on the objective of the local attention.

3. If possible, please add the ablation study by comparing the results of PrACTiS with the results of PrACTiS (but no midpoint inference), the results of PrACTiS (but no variance test), and the results of PrACTiS (but no local attention).

---

> ### Author Response · Authors · 2023-11-19
> **Response to Reviewer jRTE**
>
> Thank you for your constructive feedback. Our primary objective is to enhance the state-of-the-art TACTiS model by integrating the Perceiver IO model as its encoder. This modification enables us to tackle the previously unexplored realm of multi-modal prediction. We believe that our empirical research in this direction aligns well with the preferences of this conference's audience.
>
> We apologize for any confusion in the current manuscript and are committed to revising it to ensure greater clarity. We appreciate your input. If you have any further suggestions or specific areas of concern, please highlight them, and we will address them comprehensively in the revised version.

---

### Official Review · Reviewer_JVje · 2023-11-06

**Soundness:** 3 good
**Presentation:** 2 fair
**Contribution:** 2 fair
**Rating:** 5
**Confidence:** 2

**Summary:**

The paper presents Preceiver-Attentional Copulas (PrACTiS), a new method for time series forecasting. PrACTiS provides an efficient solution by combining Preceiver IO model with attention-based copulas. The idea of combining transformers with Copulas has been proposed in TACTiS [Drouin et al. 2022]. This paper proposes using Preciever IO for the encoder to overcome the computational cost of transformers. Experimental results are presented to validate the performance of PrACTiS.

**Strengths:**

The points of strengths include:

- The proposed method performs well compared to TACTiS and is more efficient

**Weaknesses:**

The points of weaknesses include:

1- The idea of combining transformers with copulas for forecasting has been proposed in TACTiS [Drouin et al. 2022]. The contribution seems to be limited to replacing transformers with Perceiver IO to increase efficiency which is already a known fact.

2- Some parts of the paper need more clarity such as a summary of contributions.

**Questions:**

- Could you please clarify why you did not include more baselines such as Autoformer [Wu et al. 2021]?

---

> ### Author Response · Authors · 2023-11-19
> **Response to Reviewer JVje**
>
> Thank you for providing feedback on our work. Our primary objective is to enhance the existing TACTiS model by leveraging the Perceiver architecture. We are pleased to report a substantial improvement in prediction performance for unimodal tasks, with gains of up to 40%. Notably, we successfully extend the applicability of our proposed model to multimodal tasks, achieving a significant performance boost compared to the original paper.
>
> We acknowledge the absence of a comparison with the Autoformer model, mainly designed for long-term prediction under different assumptions. In our revised version, we will address this gap by incorporating a comparative analysis with Autoformer, emphasizing the distinctions between our approach and this specific technique.
>
> We are committed to enhancing the clarity of our paper. We appreciate your insights, and the forthcoming revisions will aim to provide a more transparent motivation for our work. If you have any additional suggestions or concerns, please feel free to share them, and we will address them in our revision. Once again, thank you for your thoughtful feedback.

---

### Meta-Review · Area_Chair_iG9U · 2023-12-06

**Metareview:**

This paper presents a time series prediction model, PrACTiS (Perceiver-Attentional Copulas for Time Series). PrACTiS consists of the perceiver-based encoder and the copula-based decoder, which models the time series using a compact latent space to overcome the computational cost of transformers. Experimental results are presented to validate the performance of PrACTiS.

The studied problem is an important one. The proposed method is more efficient than TACTis. Many experiments show the competitive performance of this method against SOTA methods.

However, the novelty of this paper is limited, since it only changes the encoder and heavily extends upon TACTiS. Secondly, more experiments should be provided, such as comparison with more SOTA baselines and incomplete datasets. Thirdly, the authors should polish this paper more carefully.

**Justification For Why Not Higher Score:**

The major concern is its limited novelty.

**Justification For Why Not Lower Score:**

N/A

---

### Decision · Program_Chairs · 2024-01-16

Reject